# How Organic Substances Promote the Chemical Oxidative Degradation of Pollutants: A Mini Review

**Zhewei Hu [1], Jiaqi Shi [1,\*], Hao Yang [2], Jianbo Huang [1] and Feng Sheng [1]**

1   State Environmental Protection Key Laboratory of Soil Environmental Management and Pollution Control, Nanjing Institute of Environmental Sciences, Ministry of Ecology and Environment of China, Nanjing 210042, China; huzhewei@nies.org (Z.H.); huangjianbo@nies.org (J.H.); sf0209@126.com (F.S.)
2   College of Environment, Hohai University, Nanjing 210098, China; yhhhu588@163.com
\*   Correspondence: sjq@nies.org

**Abstract:** The promotion of pollutant oxidation degradation efficiency by adding organic catalysts has obtained widespread attention in recent years. Studies have shown that organic substances promote the process of traditional oxidation reactions by accelerating the redox cycle of transition metals, chelating transition metals, activating oxidants directly to generate reactive oxygen species such as hydroxyl and sulfate radical, or changing the electron distribution of the target pollutant. Based on the promotion of typical organic functional groups on the chemical oxidative process, a metal-organic framework has been developed and applied in the field of chemical catalytic oxidation. This manuscript reviewed the types, relative merits, and action mechanisms of common organics which promoted oxidation reactions so as to deepen the understanding of chemical oxidation mechanisms and enhance the practical application of oxidation technology.

**Keywords:** organics; chemical; oxidation; Fenton; activated; persulfate; metal-organic framework

## 1. Introduction

Chemical oxidation technology has gradually become one of the mainstream technologies applied in water body and contaminated site restoration engineering due to its good treatment effect and low application cost. Commonly used chemical oxidants include potassium, ozone, Fenton and Fenton-like reagents, activated persulfate (PDS), and etc. The traditional chemical oxidation technology has some deficiencies, such as low oxidation efficiency, pH limitation, and sludge accumulation caused by iron salt [1–3]. Therefore, a significant amount of research has begun to focus on exploring the technical means to improve the efficiency of chemical oxidation or broaden the pH scope of application. For example, the $Fe^{3+}$, $Cu^{2+}$, $Co^{2+}$, and $Mn^{2+}$ were used to replace $Fe^{2+}$ to active $H_2O_2$, which were feasible for broadening the oxidative pH but would often decrease the oxidation rate [4,5], and solid $Fe^0$ or ores could also activate $H_2O_2$ or PDS in near-neutral condition [6,7]. In recent years, more and more studies have revealed that the introduction of organic substances into the system can serve such purposes, and some oxidation intermediates of organic pollutants could also self-accelerate the reaction [8,9].

The organic substances which could improve the efficiency of chemical oxidation mainly existed in three forms: organic small molecular form itself, organic chelant, and metal-organic frameworks (MOFs) material. The recent research advances on the types of organic substances, their environmental affinity, and action mechanisms that can effectively improve the efficiency of oxidative degradation of pollutants were summarized in this manuscript, which means to provide a reference for the further study on chemical oxidation kinetics, application conditions, and mechanisms.

## 2. Reaction Acceleration by Organic Small Molecular Substances

### 2.1. Accelerated the Metal Redox Cycle

Huang et al. showed that in the $Fe^{2+}/Fe^{3+}$ cycle, $Fe^{2+}$ regeneration is the oxidation rate control step [10]. Any method to improve the $Fe^{3+}$ reduction efficiency will accelerate the formation of hydroxyl radicals ($\bullet OH$), which is also the key to affect the Fenton/Fenton-like oxidation rate [11,12]. Compounds rich in nucleophilic groups such as carboxyl, hydroxyl, carbonyl, and amino groups can accelerate the $Fe^{2+}/Fe^{3+}$ cycle due to their strong reducibility, thereby accelerating the generation of free radicals. Subramanian and Madras [13] found that thioglycolic acid can effectively improve the $Fe^{2+}/Fe^{3+}$ cycle efficiency in the Fenton system under near-neutral pH conditions, and accelerate the degradation efficiency of organic pollutants. Moreover, Chen et al. showed that hydroxylamine can enhance the production of $\bullet OH$ by promoting $Fe^{3+}$ reduction, and it is still effective when the pH is 5.7. Hydroxylamine was finally oxidized to $NO_3^-$ and $N_2O$ in the system [14].

Advanced oxidation technology based on sulfate radical ($SO_4^-\bullet$) has similar characteristics [15,16]. For example, Yang et al. found that ascorbic acid (AA) can promote the oxidation effect of the $Fe^{2+}/PDS$ system at a pH range of 2.0 to 6.2 [17]. Garcia et al. studied the difference in the effect of tartaric acid (TA) and hydroxylamine hydrochloride added into the $Fe^{2+}/PDS$ system on the degradation process of bisphenol A [18,19]; the results showed that they both accelerated the efficiency of free radical generation and broadened the pH range of the reaction. The difference is that hydroxylamine hydrochloride only has a short-term promotion effect, while TA can promote the oxidation effect for a long time. The $Fe^{2+}$ activates PDS to generate $SO_4^-\bullet$, which reacts with TA to generate organic compound radicals ($R\bullet$), then $Fe^{3+}$ reacts with $R\bullet$ slowly to generate $Fe^{2+}$, forming a chain reaction which promotes the oxidation reaction for a long time. This is consistent with the results of Minisci et al. and Liang et al. [20,21].

Phenolic and quinone compounds have also been found to have the effect of accelerating Fenton/PDS oxidation. Xiao et al. found that phenolic compounds accelerated the Fenton-like degradation of dimethyl phthalate (DMP), with an increased efficiency of $H_2O_2$ utilization. The effect of phenolic compounds on the degradation of DMP followed the order: catechol $\approx$ hydroquinone > resorcinol > phenol, which could be attributed to the interaction between quinone-like substances and iron ions. Hydroquinone-like substances accelerated the Fe(III)/(II) redox cycle. The formation of iron complexes between catechol-like substances and iron ions facilitated the release of $H^+$ and regeneration of Fe(II) [22]. Phenols and quinones are also typical intermediate oxidation products of aromatic compounds [8]. Jiang et al. [9,23] found that hydroquinone and p-benzoquinone, which were the oxidation intermediates of phenol and nitrobenzene, can promote the decomposition of Fe(III)–hydroperoxy complexes, and promote the conversion of Fe(II)/Fe(III). Therefore, the degradation of such aromatic compounds has an autocatalytic effect.

The promotion is also effective in heterogeneous oxidation system. Sun et al. found that AA can significantly increase the $Fe^{2+}/Fe^{3+}$ cycle on the surface of magnetite ($Fe_3O_4$), thereby accelerating the oxidation efficiency of the $Fe_3O_4/H_2O_2$ system on alachlor [24]. A Fenton-like system with $MnO_x$-$Fe_3O_4$/biochar composite (FeMn/biochar) was constructed for pollutant degradation. Five well-characterized reducing agents (sodium borohydride (SBH), sodium thiosulfate (STS), AA, hydroxylamine, and oxalic acid (OA)) were added respectively to investigate their performance to the system. The results revealed that only OA and hydroxylamine obviously enhanced the catalytic capacity of the Fenton-like process and HA increased ciprofloxacin degradation efficiency from 38.2% to 92.8%, well in agreement with the accelerated Fe(III/II) cycle and Mn(III/II) cycle in the $Fe/Mn/biochar$-$H_2O_2$-HA system. The accelerated metal redox cycle could enhance the decomposition of $H_2O_2$ into $\bullet OH$ and $\bullet O_2^-$, which were the main reactive oxygen species responsible for ciprofloxacin degradation [25]. Sang et al. found that proper addition of hydroxylamine greatly promoted the degradation of sulfamethoxazole in each of the $\alpha$-$Fe_2O_3$/peroxomonosulfate (PMS), $Co_2O_3$/PMS, and CuO/PMS systems. The results suggested that $\bullet OH$ radical was the dominating reactive oxygen species in the

$\alpha$-Fe$_2$O$_3$/hydroxylamine/PMS and Co$_2$O$_3$/hydroxylamine/PMS systems, while in the CuO/hydroxylamine/PMS system, the activated-PMS on the surface of CuO was concluded to be responsible for SMZ degradation instead of free radicals [18].

## 2.2. Activated H$_2$O$_2$ or PDS to Construct Advanced Oxidation System

In addition to accelerating the transition metal cycle in the advanced oxidation system, organic matter can also directly activate the oxidant to generate free radicals to build an advanced oxidation system.

### 2.2.1. Organic Matters Activate H$_2$O$_2$

Studies have shown that organic matter can directly activate H$_2$O$_2$ to produce oxidizing active substances, such as $\bullet$OH, O$^{2-}$ and $^1$O$_2$, which strengthen the oxidation capacity. The activation is generally related to the organic free radicals generated during the reaction (see Figure 1) [26,27].

RO　　$\bullet$ OH/SO$_4^-\bullet$　　Target pollutant

Organic such as ROH　　RO $\bullet$　　H$_2$O$_2$/Na$_2$S$_2$O$_8$

**Figure 1.** The mechanism of H$_2$O$_2$/PDS activation by organics.

In addition to being used as a promoter to improve oxidation efficiency, hydroxylamine can also directly activate H$_2$O$_2$ to form an advanced oxidation system. Chen et al. proposed that hydroxylamine activating H$_2$O$_2$ to produce $\bullet$OH may be divided into two steps: the first step is hydroxylamine ions to activate H$_2$O$_2$ to produce $\bullet$OH [27]; the second step is to produce $\bullet$OH by the reaction of H$_2$O$_2$ with the protonated amino radical generated in the first step. They also proposed that the reaction between hydroxylamine and H$_2$O$_2$ to generate $\bullet$OH may be related to the -OH group in hydroxylamine.

The quinone structure can also activate H$_2$O$_2$ and promote the generation of $\bullet$OH. Zhu et al. studied the production mechanism of $\bullet$OH during the activation of H$_2$O$_2$ by haloquinone, and the results showed that $\bullet$OH is generated by tetrachloro-1,4-benzoquinone and H$_2$O$_2$ through a mechanism that has nothing to do with metals: the nucleophilic attack of H$_2$O$_2$ on tetrachloro-1,4-benzoquinone forms a trichlorohydroperoxy-1,4-benzoquinone (TrCBQ-OOH) intermediate, which is further cracked to produce $\bullet$OH [28,29].

### 2.2.2. Organic Matter Activates PDS

With the rapid development of PDS activation technology, the new type of activation technology for organic activation of PDS has received widespread attention. Studies have shown that organic compounds such as quinone compounds, AA, hydroxylamine, phenols, quercetin, and surfactants can activate PDS.

Fang et al. found that both quinone compounds and humic acid can effectively activate PDS to degrade 2,4,4'-trichlorobiphenyl [30]. Its degradation rate could reach 88% in the p-benzoquinone/PDS system, while the degradation rates in a single PDS or p-benzoquinone system were only 20% and 9%, respectively. Zhang et al. explored the effects of different types of anthraquinone dissolved organic matter on the degradation of Rhodamine B by PDS [31]. Since anthraquinone-dissolved organic matter contains an oxidation-sensitive functional group structure, it not only transfers electrons in the PDS activation reaction [32], but also generates reductive semiquinone radicals during the activation process, which reduces S$_2$O$_8^{2-}$ to SO$_4^-\bullet$ and SO$_4^{2-}$ and can significantly enhance PDS oxidation capacity [33].

When studying the activation of PDS by AA, Hou et al. used the AA/PDS system to degrade atrazine. Compared with a single PDS oxidation system, after AA, SO$_4^-\bullet$ and $\bullet$OH are generated in the system, the degradation rate of atrazine is increased by 29 times [34]. Cao et al. showed that when the pH of the AA/PDS system is between 3.5

and 12.5, PDS is mainly activated by AA. When pH > 12.5, PDS is mainly activated by alkali [35].

The activation effect of phenols on PDS is mainly achieved by phenates. When the pH is 8.3, pentachlorophenol degrades pollutants by reducing PDS to generate active free radicals [36]. Quercetin (QCR) is a flavonoid polyphenol organic compound whose structure lacks electron delocalization and can release electrons to activate PDS, and then produce $SO_4^-\bullet$ and $\bullet OH$. When the pH is 13, compared with a single PDS system, the QCR/PDS system can effectively degrade 1,1-dichloroethane, 1,2-dichloroethane, 1,2-dichloropropane, and dibromomethane [37].

Surfactants are usually used for the desorption of pollutants and the dissolution of non-aqueous liquids in in-situ chemical oxidation processes. The results of studies have shown that anionic, nonionic, and cationic surfactants (docusate sodium, polyethylene glycol 400, and N-tallow propylene diamine polyoxyethylene ether) can effectively activate PDS [38]. Among them, the cationic surfactant N-tallow-based propylene diamine polyoxyethylene ether shows the strongest activation effect. It can generate $\bullet OH$ under alkaline conditions, and can generate reducing or nucleophilic groups (superoxide radicals, hydroperoxide anions, alkyl radicals, etc.) under acidic and alkaline conditions.

### 2.3. Other Effects

Except for Fenton, Fenton-like, and PDS, organic small molecular substances can also enhance the oxidizing ability of other oxidants. Yang et al. showed that the removal efficiency of phenol and bisphenol A (BPA) permanganate has a synergistic effect under weak acid conditions (pH 4.0~6.0) [39]. Among them, the removal effect of phenol increased with the increase of the initial concentration of BPA, but decreased with the pH. However, under weakly alkaline (pH 7.5~8.5) conditions, the two had a competitive effect on the degradation of permanganate. That is, the degradation of phenol was inhibited in the presence of BPA, but the degradation efficiency of BPA was slightly improved in the presence of phenol. The study speculated that the reason for the synergy is that bisphenol A induced the production of manganese oxide in the system, and the reason for the competition may be the formation of reactive manganese intermediates $Mn^{5+}$ or $Mn^{4+}$ in the oxidation system.

Degradation of pollutants by ozonation in the presence of hydroxylamine has been investigated in previous studies [40,41]. The results showed that the degradation rate could be improved obviously in the presence of HA, which was attributed to the production of $\bullet OH$ and singlet oxygen.

### 3. The Promotion of Organic Chelating Agents

Organic complexing agents can effectively prevent the precipitation of transition metals in a non-acidic environment, and the formed coordination field often affects the redox characteristics of $Fe^{2+}/Fe^{3+}$, thereby promoting the oxidation effect of the Fenton (Fenton-like) system [42,43]. The main action mechanism of organic small molecular substances and complexing agents are displayed in Figure 2. Commonly used organic complexing agents include humic substances, carboxylic acid compounds, and amino carboxylic acid compounds.

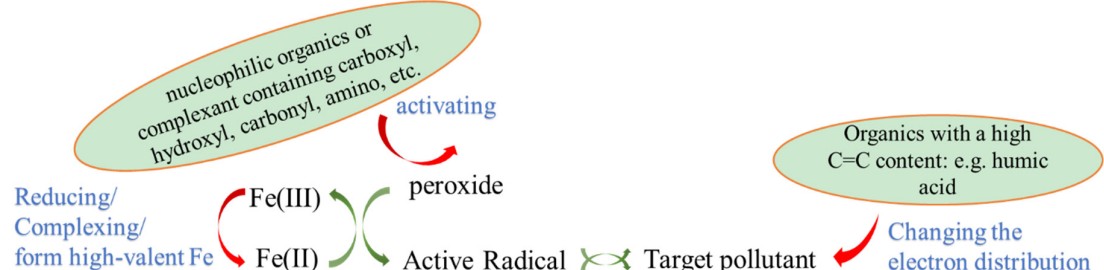

**Figure 2.** The main action mechanism of small molecular substances and chelator in Fe-mediated systems.

### 3.1. Humic Substances

Humus widely exists in natural water bodies, soils, and sediments [44]. Metal ions can form chelates with humic acid (HA) or fulvic acid (FA) by binding to carboxyl groups, phenolic groups, and nitrogen-containing sites [45], and HAs and FAs can effectively reduce $Fe^{3+}$, thereby accelerating the Fenton system oxidation process [46,47]. Research results show that different concentrations of humic acid in the pH range of 5~7 often have a good effect on Fenton (Fenton-like) reaction [42,48]. However, when the pH is low, humic acid often has no obvious effect on the Fenton (Fenton-like) reaction, and even exerts a certain inhibitory effect. For example, Lipczynska-Kochany and Kochany found that adding 3000 mg $L^{-1}$ humate to the Fenton system of pH 7 could greatly improve the removal efficiency of pollutants. However, when the pH was 3.5, the addition of humic acid salt actually inhibited the degradation [49]. FA presents similar performance [49,50]. Lindsey et al. showed that the inhibitory effect of humus on the degradation of pollutants under acidic pH conditions is related to the hydrophobicity of the substrate [51].

Humic acid can also promote the oxidation efficiency of permanganate and ozone. He et al. found that humic acid could promote the removal of phenol by potassium permanganate under the condition of pH 4~8, while it would inhibit the oxidation of phenol by potassium permanganate under the condition of pH 9~10 [52]. Potassium permanganate is an electrophilic reagent, and the oxidation rate increases with the increase of the electron cloud density on the aromatic ring of the target pollutant [53]. The aromatic ring structure in humic acid forms a $\pi$-$\pi$ interaction with phenol, which increases the electron cloud density of phenol, thereby promoting the oxidation capacity of potassium permanganate [26,54]. The macromolecular humic acid has a high C = C content, and the $\pi$-$\pi$ interaction has a good positive correlation with the C = C content. Therefore, the macromolecular humic acid further promotes the oxidation ability of potassium permanganate. Yang et al. found that adding low concentration (such as 1 mg $L^{-1}$) humus to the manganese-catalyzed ozone oxidation system can help improve the degradation rate of atrazine [55]. However, when the concentration of humus is further increased, the degradation of atrazine is inhibited. The reason is that the low concentration of humic acid helps to stimulate the formation of •OH in the system, while the quenching of free radicals under the condition of high concentration of humic acid plays a leading role.

### 3.2. Carboxylic Acid

Molecular molar ratio and pH are important factors that affect the catalytic effect of Fe-carboxylic acid chelating agents. Generally, when the pH is in the range of acidic to neutral, the Fe-carboxylic acid chelating agent has a better catalytic effect. Citric acid (CA) is a commonly used carboxylic acid Fenton reaction complexing agent. Studies have shown that when the molar ratio of Fe and CA molecules is 1:1, there are three main forms, namely, [Fe(Cit)]°, [Fe(Cit)]⁺, and [Fe(Cit)(OH)] [56,57]. Studies have also reported the formation of a complex with a molar ratio of Fe and CA of 2:2 [58]. Lewis et al. showed that the Fenton reaction modified by CA chelation can effectively reduce the usage amount of $Fe^{2+}$ under near-neutral (pH 6–7) conditions [59]. In addition, when the molar ratio of CA and Fe molecules is between 1:1 and 4:1, increasing the molar ratio of molecules will reduce the decomposition efficiency of $H_2O_2$. In addition, Li et al. pointed out that even when the pH is greater than 5, the Fenton system with CA as a chelating agent can still efficiently degrade 2,4,6-trichlorophenol. However, when pH > 8, a large amount of iron precipitation is formed [60]. Trovo et al. showed that CA can extend the application range of the photo-Fenton system to neutral pH conditions, and the degradation rate heavily depends on the initial concentration of citrate [61]. When the pH is 5–8, the degradation efficiency of diclofenac (DCF) in the photofenton method decreases with the increase of pH.

Except for CA, in recent years, some new carboxylic acid natural organic complexing agents have also been found to not only promote the $Fe^{2+}/Fe^{3+}$ cycle, but also effectively prevent the precipitation of iron ions. For example, Qin et al. found that adding proto-catechuic acid (PCA) to the $Fe^{3+}/H_2O_2$ Fenton-like system could effectively increase the

degradation rate of alachlor when the pH was less than 7, but the degradation efficiency decreased with the increase of pH; when pH $\geq$ 7, PCA no longer increased the degradation rate of alachlor [62]. Ren et al. showed that rosmarinic acid (RA) could greatly enhance the oxidation efficiency of 2,4-dichlorophenol in the $Fe^{3+}/H_2O_2$ system when the pH is was the range of 3–6 [63]. However, when the pH increased to 6.4 and 7.2, the promotion effect gradually weakened. Generally, both the target pollutant and the organic complexing agent can be effectively mineralized in the Fenton system, and this type of organic complexing agent has good environmental friendliness [62,63].

### 3.3. Amino Carboxylic Acid

Because of its strong complexing ability, amino carboxylic acid can prevent iron precipitation even under neutral pH conditions, and it is also a common organic complexing agent [10,64]. Ethylenediamine tetraacetic acid (EDTA) is the most commonly used aminocarboxylic acid complexing agent. The 1:1 organic ligand $Fe^{2+}$ (EDTA) formed by EDTA-2Na and $Fe^{2+}$ can not only extend the time of $Fe^{2+}$ participating in the Fenton reaction, but also activate the dissolved oxygen in the system and spontaneously generate $H_2O_2$ [65]. Brian et al. showed that when the molar ratio of EDTA-2Na to $Fe^{2+}$ was 1:1, the degradation effect of benzene and 1,2-dichlorobenzene was the best [66]. When the concentration of EDTA-2Na is further increased and exceeds the requirement for complexation, it will consume $\bullet$OH by itself to reduce the degradation rate. In the EDTA-$Fe^{3+}$-$H_2O_2$ system, as the molar ratio of EDTA: $Fe^{3+}$ (range 1:1~5:1) increases, the decolorization efficiency of malachite green gradually increases. The degradation mechanism does not follow the simple hydroxyl radical mechanism, and the intermediate valence iron (four or five valence) that exists at the same time plays a major role in oxidation [67]. In recent years, EDTA has also been found to promote the activation of $H_2O_2$ by heterogeneous oxidation systems such as zero-valent iron or iron oxides [68–70]. However, EDTA has a strong capacity to chelate heavy metals and poor biodegradability, which may exert an adverse effect on the environment [71,72]. Therefore, in recent years, the search for biodegradable EDTA substitutes has gradually attracted the attention of researchers [73–75], such as N, N′-ethylenediamine disuccinic acid (EDDS), which is a structural isomer of EDTA and can exist in [S,S′], [S,R/R,S], and [R,R′] configurations (see Figure 3). Among them, the [S,S] configuration can be quickly and completely mineralized, and the other two configurations can be partially biodegraded [76]. The isomeric mixture of EDDS contains statistically 25% of (S,S)-EDDS, 25% of (R,R)-EDDS, and 50% of (R,S)/(S, R)-EDDS. Orama et al. proposed that the suitable pH range of EDDS as a $Fe^{3+}$ chelating agent is 3–9 [77]. When pH $\leq$ 7, $Fe^{3+}$-EDDS complex mainly exists in the form of $Fe^{3+}$-EDDS-; when pH > 7, the $Fe^{3+}$-EDDS complex mainly exists in the form of $Fe(OH)EDDS^{2-}$ and $Fe(OH)_2EDDS^{3-}$. Huang et al. found that in the Fenton reaction driven by EDDS, under neutral or alkaline conditions, due to the generation of $\bullet HO_2$ or $\bullet O_2^-$ radicals and the presence of various forms of complex iron, the oxidation efficiency was much higher than that of acidic conditions [8]. EDDS can not only keep iron in a soluble form, but also promote the generation of superoxide radicals, thereby promoting the generation of $Fe^{2+}$ and $\bullet$OH. Nitrilotriacetic acid (NTA) is also an amino carboxylic acid complexing agent that has been frequently studied in recent years. Sun and Pignatello tested 50 kinds of structurally diverse organic and inorganic polydentate chelators for their ability to solubilize Fe(III) at pH 6 and catalyzed the oxidation of 0.1 mM 2,4-dichlorophenoxyacetic acid by 10 mM $H_2O_2$ in aerated aqueous solution [78]. The result showed that NTA was one of the most active chelates that promote the degradation. The reactivity of the $Fe^{3+}$-NTA system would not be affected by excessive NTA [79]. There is only one N atom in the NTA molecule, and it could be degraded by microorganisms under hypoxic conditions, which has a small risk of causing environmental problems [80].

**Figure 3.** Chemical structure of [S,S], [S,R/R,S], and [R,R] configurations of EDDS.

Moreover, some other N-containing organic complexing agents have also been proven to have better $Fe^{3+}$ complexing and the capacity to promote $H_2O_2$ decomposition. In addition to the production of hydroxyl radicals, the oxidation process is often accompanied by the production of high-valent iron compounds [78,81].

The promoting effect of organic matters on the Fenton-like system is also related to the type of transition metal. For example, Ghiselli et al. found that at pH 5.5, EDTA, NTA, and CA could inhibit the degradation of organic pollutants in the Fenton-like reaction catalyzed by copper [82], while TA could promote the degradation. The reason may be that EDTA, NTA, and CA have strong complexing capacity with $Cu^{2+}$ and high stability constants, which prevented the interaction between the effective sites of Cu ions and $H_2O_2$. TA itself has strong reducibility and can reduce $Cu^{2+}$ existing in a complex state to $Cu^+$. The reaction of $Cu^+$ with $H_2O_2$ can increase the production of ●OH and promote the oxidative degradation of organic pollutants.

The characteristics of different types of chelating agents are listed in Table 1, and the application of typical organic small molecular substances and chelating agents in promoting oxidation reaction is summarized in Table 2.

**Table 1.** The merits and demerits of different chelators.

| Chelator | Merit | Demerit |
|---|---|---|
| Humus | Ubiquitous in nature | No obvious influence and even inhibition when pH is low |
| Carboxylic acid | Suitable for acid to neutral range; Chelating agent can effectively degrade | Affected by pH and molar ratio to transition metal |
| Amino carboxylic acid | The time of $Fe^{2+}$ participation in Fenton reaction can be extended; It can activate the dissolved oxygen in the system and produce $H_2O_2$ spontaneously; The effect is better under neutral neutral or alkaline conditions | The molar ratio of transition metal has influence on the promoting effect; Environmental degradation is poor |

**Table 2.** Chemical oxidation degradation of pollutants by organics.

| Organics Class | Organic Substance | Structure | Oxidant | Reaction pH | Target Pollutant | Source |
|---|---|---|---|---|---|---|
| Phenols | Hydroquinone |  | $H_2O_2$ (10 mmol $L^{-1}$), $Fe^{3+}$ (0.5 mmol $L^{-1}$) and hydroquinone (0.1 mmol $L^{-1}$) | 3.1~3.2 | DMP (1 mmol $L^{-1}$) | [22] |
| | Pentachlorophenol |  | $Na_2S_2O_8$ (0.5 mol $L^{-1}$) and pentachlorophenol (1 mmol $L^{-1}$) | 6.5~10.5 | Nitrobenzene (1 mmol $L^{-1}$) | [36] |
| | AA |  | $H_2O_2$ (1 mmol $L^{-1}$), AA (0.5 mmol $L^{-1}$) and $Fe_3O_4$ (1 g $L^{-1}$) | 4 | Alachlor (20 mg $L^{-1}$) | [24] |
| | | | $Na_2S_2O_8$ (40 mmol $L^{-1}$) and AA (1.0 mmol $L^{-1}$) | 7.2 | Pentachlorophenol (10 mg $L^{-1}$) | [35] |
| Humus | Humic Acid | / | $H_2O_2$ (130 mmol $L^{-1}$), $Fe^{2+}$ (30 µmol $L^{-1}$) and HA (50~100 mg $L^{-1}$) | 5~7 | Benzene (25 µmol $L^{-1}$) | [42] |
| | | | $H_2O_2$ (50 mmol $L^{-1}$), $Fe^{2+}$ (5 mmol $L^{-1}$) and HA (10 mg $L^{-1}$) | 6.5 | 15 organic compounds | [48] |
| Carboxylic Acid Compounds | CA |  | $H_2O_2$ (50 mmol $L^{-1}$), $Fe^{2+}$ (10 mmol $L^{-1}$) and CA (10 mmol $L^{-1}$) | 5~7 | 2,4,6–trichlorophenol (1.5 mmol $L^{-1}$) | [60] |
| | Gallic Acid |  | $H_2O_2$ (8 mmol $L^{-1}$) and $Fe^{3+}$ (0.1 mmol $L^{-1}$) | 3.6 | gallic acid (0.11 mmol $L^{-1}$) | [62] |
| Amino Carboxylic Acids | EDTA |  | $Na_2S_2O_8$ (5 mmol $L^{-1}$), $Fe^0$ (1.0 g $L^{-1}$) and EDTA (1 mmol $L^{-1}$) | 6.0 | Reactive Green 19 (0.05 mmol $L^{-1}$) | [69] |
| | EDDS | a mixture consisted of different configurations | $H_2O_2$ (5 mol $L^{-1}$) and $Fe^{3+}$-EDDS (1 mol $L^{-1}$) | 6.2 | bisphenol A (20 µmol $L^{-1}$) | [10] |
| | NTA |  | $H_2O_2$ (100 mmol $L^{-1}$), $Fe_3O_4$ (1.0 g $L^{-1}$) and NTA (0.5 mmol $L^{-1}$) | 7 | Carbamazepine (63.5 µmol $L^{-1}$) | [73] |
| Others | Hydroxylamine |  | $H_2O_2$ (0.4 mmol $L^{-1}$), $Fe^{2+}$ (10.0 µmol $L^{-1}$) and $NH_2OH$ (0.4 mmol $L^{-1}$) | 2.0~5.7 | benzoic acid (40.0 µmol $L^{-1}$) | [14] |
| | 1,4-benzoquinone |  | Persulfate (5 mmol $L^{-1}$) | 7.4 | PCB 28 (0.5 mg $L^{-1}$) | [30] |

## 4. Metal-Organic Framework Materials

In recent years, solid-phase catalytic materials have attracted extensive attention. The heterogeneous oxidation system composed of solid-phase catalytic materials and oxidants overcomes the disadvantages of homogeneous Fenton catalyst, such as difficult separation and repeated use, easy formation of iron sludge by $Fe^{2+}$, increase of color in water, and narrow reaction pH range. Among them, metal-organic frameworks (MOFs) which have intramolecular voids formed by self-assembly of organic ligands and metal ions or clusters through coordination bonds [83–86], have a higher catalytic activity than traditional solid catalytic material and have been used in advanced oxidation systems. MOFs combines the excellent properties of organic and inorganic substances, so its effectiveness exceeds the performance of simple mixing [87,88]. Due to the diversity of metal ion species and the selection of organic ligands, MOFs materials have many advantages: high specific surface area, functional groups such as $-NH_2$, $-OH$, and $-COOH$ which can be introduced to the pore surface of MOFs material, high thermal and chemical stability, and etc. The performance can be adjusted through the synergistic effect of metal ions and organic functional groups, and post-synthetic modification can be performed to make it have special properties [89]. A typical mechanism for pollutant degradation catalyzed by MOF is shown in Figure 4.

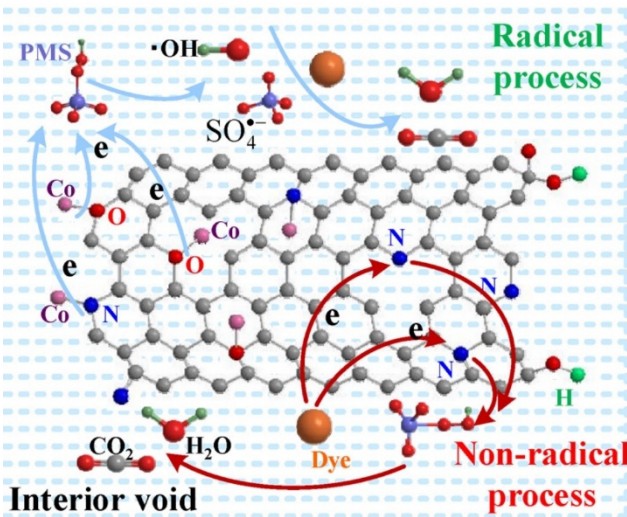

**Figure 4.** Mechanisms for dye degradation by the Co@N-C/PMS system. Reproduced with permission from Yao [90].

### 4.1. Fe-Based MOFs Catalytic Materials

Iron-based MOFs are porous materials assembled from iron ions or iron clusters and organic ligands, which can promote the $Fe^{3+}/Fe^{2+}$ conversion rate in MOFs by means of metal doping and organic ligand modification and improve the heterogeneity catalytic $H_2O_2$ degradation efficiency of pollutants [91]. Fe-based MOFs were widely studied because Fe was eco-friendly and cost efficient. So far, a large number of iron-based MOFs using different types of organic ligands have been found. For example, MIL-53 (Fe), MIL-88B(Fe), and MIL-100 (Fe) have been reported as Fenton-like catalysts to degrade pollutants [92–94]. Li et al. prepared $Fe^{2+}$-containing MOFs using 2, 2′-bipyridine-5,5′-dihydroxy acid as ligand [95]. The material activated $H_2O_2$ to degrade organic pollutants in near neutral conditions, and it had an excellent activity and stability. A high $H_2O_2$ utilization efficiency was also achieved. Lv et al. used $Fe^{2+}$@MIL-100(Fe) Fenton catalyst to study the effect of MB degradation. The experimental results showed that $Fe^{2+}$@MIL-100 (Fe) demonstrated the highest Fenton catalytic ability compared with MIL-100 (Fe) and $Fe_2O_3$ [94]. The $Fe^{2+}$ and $Fe^{3+}$ ions in $Fe^{2+}$@MIL-100 (Fe) have a synergistic effect on the production of •OH, thereby promoting oxidation efficiency.

*4.2. Copper- and Cobalt-Based MOFs Catalytic Materials*

Other than Fe-based MOF, other metals such as copper and cobalt were also found to have the ability to catalyze $H_2O_2$ to generate •OH [96,97]. Because of the appropriate price and wide distribution, Cu was usually used in catalytic material. Lyn et al. synthesized Cu-doped mesoporous silica microspheres (Cu-MSM) by the hydrothermal method, and the Fenton-like process catalyzed by Cu-MSMs showed excellent performance in the degradation of phenytoin sodium (PHT) and diphenhydramine (DP). The reaction mechanism is that $H_2O_2$ was converted to •OH by the framework $Cu^+$ in Cu-MSMs, and $Cu^+$ was simultaneously oxidized to $Cu^{2+}$. The generated •OH could cause the decomposition of PHT and DP, while the generated phenolic intermediates could be adsorbed on the surface of Cu-MSMs, complexed with the framework $Cu^{2+}$ and form copper complexes. The complex can interact with $H_2O_2$ and promote $Cu^{2+}$ reduction, accelerate the $Cu^+/Cu^{2+}$ circulation, generate •OH more efficiently, and thus promote the degradation of organic pollutants.

Cobalt ion is also a common heterogeneous Fenton oxidant, which can be used to catalyze oxidants such as $H_2O_2$, PDS, and peroxymonosulfate [91]. Co has a higher catalytic activity than Fe and Cu. For example, Racles et al. synthesized two kinds of MOFs containing Cu and Co respectively at room temperature, and added them to $H_2O_2$ to degrade the azo dye congo red (CR) [98]. The results showed that the degradation efficiency of the cobalt-based MOF material to CR reached 90% after reaction for 30 min, which was greater than that of the copper-based MOFs material. This indicates that to a certain extent the catalytic capacity of the cobalt-based MOFs material is greater than that of the copper-based MOFs material.

*4.3. Multi-Core MOFs Catalytic Materials*

In recent years, MOFs doped with multiple metals have been widely concerned [99,100]. Li et al. studied and synthesized Fe-Co Prussian blue complexes as photo-Fenton catalysts, and this material has high degradation efficiency for rhodamine B under the conditions of pH 3.0–8.5 [101]. During the entire photo-Fenton reaction process, $H_2O_2$ molecules replaced water molecules coordinated with Fe, and the resulting $Fe^{2+}$–peroxide complex could generate •OH. At the same time, $Fe^{3+}$ in the catalyst was reduced by $H_2O_2$, and the generated HOO• and •OH reacted to form $^1O_2$, which directly participated in the degradation of rhodamine B. Wang et al. showed that under the condition of pH 5, the removal rate of 20 mg·$L^{-1}$ MB by the MIL-101(Fe,Cu)/$H_2O_2$ system was 100% when the reaction time was 20 min [102]. Compared with MIL-101(Fe)/$H_2O_2$ and $H_2O_2$ alone, it increased by 43.1% and 88.9%, respectively. This is because $Cu^{2+}$ doping introduces new active sites, where $Cu^{2+}/Cu^+$ can synergize with $Fe^{3+}/Fe^{2+}$ cycles to produce more •OH to improve the Fenton-like degradation effect.

## 5. Conclusions and Prospects

The oxidative degradation of pollutants in the water can be enhanced by adding organic substances or preparing solid materials with organic substances. Organics containing nucleophilic functional groups such as carboxyl, quinone, hydroxyl, and amino groups can effectively accelerate the generation of free radicals and broaden the pH range in Fenton, Fenton-like, and activated PDS systems by promoting the reduction of transition metals, complexing transition metals, or inhibiting the hydrolysis of transition metals. Some quinones, phenols and carboxylic acids can even directly activate $H_2O_2$/PDS to generate free radicals. These organic substances can also improve the oxidation efficiency of permanganate by changing the valence state of manganese. In addition, based on the characteristics of organic compounds which promote chemical oxidation reactions, MOFs materials are created to improve the oxidation effect of the advanced oxidation system, which constructed a heterogeneous system to release the oxidizing species steadily or re-utilize the catalysts.

However, at present, except traditional complexing agents such as CA and EDTA which are often used to enhance the oxidation effect in actual water bodies, most organics

and MOFs materials are applied only in the laboratories. As a result, for future development of organics or organic-containing material in real environmental applications, more efforts can be made on the following aspects: (1) to decrease the acquisition cost of the organic materials; (2) to investigate their environmental risks; and (3) to clarify whether the organic matter unremoved during the chemical oxidation can offer available carbon for further microbiological deterioration of pollutants.

**Author Contributions:** Z.H. wrote the paper; J.S. conceived and designed the paper; H.Y., J.H. and F.S. collected and analyzed the data. All authors have read and agreed to the published version of the manuscript.

**Funding:** This research was funded by the National Key Research and Development Program of China, grant number 2018YFC1801100; National Natural Science Foundation of China, grant number 21707041.

**Institutional Review Board Statement:** Not applicable.

**Informed Consent Statement:** Not applicable.

**Data Availability Statement:** The data that support the findings of this study are available upon request from the authors.

**Conflicts of Interest:** The authors declare no conflict of interest.

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
