# Peer review of "How Organic Substances Promote the Chemical Oxidative Degradation of Pollutants: A Mini Review"

_sustainability, doi:10.3390/su131910993_

Round 1

Reviewer 1 Report

This manuscript reviews the types, relative merits and action mechanisms of common organics which promoted oxidation reactions. Many works have been listed as examples to investigate how organic substances promote the chemical oxidative degradation of pollutants, and many literatures about that has been cited. The problems are provided below followed by comments.

Problems:

  • The organization and the logic of the manuscript need to be improved significantly. Line 37-40: The authors indicated that “The promotion effect of organic substances on chemical oxidation is mainly achieved by the following ways: organic matter substances were mixed into the reaction solution and reacted with the catalyst or target chemical to accelerate the oxidation; H2O2/PDS were activated by organics to construct the advanced oxidation system; or the metal-organic frameworks (MOFs) material was constructed to catalyze the reaction, and so on.” However, it seems that “organic matter substances were mixed into the reaction solution and reacted with the catalyst or target chemical to accelerate the oxidation” includes “the metal-organic frameworks (MOFs) material was constructed to catalyze the reaction”.

  • A lot of work by researchers were listed and many organic substances promoting the chemical oxidative degradation of pollutants were enumerated in this manuscript. However, this manuscript looks like a report rather than a review. What were the roles of those organic substances, and how they can affect the reactions? The discussion about that in every section was unclear, please clarify.

Other Comments:

  • The manuscript needs to be revised by a native speaker. The grammatical problems and the tense used in the manuscript should be especially paid attention to.
  • Line 248: " H2O2/PDS"->" H2O2 or PDS "?
  • Line 252 and 254: Pay attention to superscripts and subscripts

Author Response

Dear Editors and Reviewers:

Thank you for your letter and for the reviewers' comments concerning our manuscript entitled “How Organic Substances Promote the Chemical Oxidative Degradation of Pollutants: A Mini Review” (No.sustainability-1352288). Those comments are all valuable and very helpful for revising and improving our paper, as well as the important guiding significance to our researches. We have studied comments carefully and have made correction which we hope meet with approval. Besides, our current manuscript followed the journal formatting guidelines of Sustainability. Revised portion are marked in red throughout the revised manuscript. The main corrections in the paper and the Responses to the reviewer’s comments are as flowing:

NOTE: All the Page and Line numbers where revisions were made refer to the Manuscript and Highlight with marked changes (Manuscript_revised version).docx. The Manuscript_Clean Version was the same version of the Manuscript_revised version with cleaned from all the marks.

Responses to the Reviewer #1’s comments:

1)The organization and the logic of the manuscript need to be improved significantly. Line 37-40: The authors indicated that “The promotion effect of organic substances on chemical oxidation is mainly achieved by the following ways: organic matter substances were mixed into the reaction solution and reacted with the catalyst or target chemical to accelerate the oxidation; H2O2/PDS were activated by organics to construct the advanced oxidation system; or the metal-organic frameworks (MOFs) material was constructed to catalyze the reaction, and so on.” However, it seems that “organic matter substances were mixed into the reaction solution and reacted with the catalyst or target chemical to accelerate the oxidation” includes “the metal-organic frameworks (MOFs) material was constructed to catalyze the reaction”.

Response: Thank you very much for your comments and suggestions to revised the manuscript. We revise Line 37-40 to “The organic substances which could improve the efficiency of chemical oxidation mainly existed in three forms: organic small molecular itself, organic chelant, and met-al-organic frameworks (MOFs) material.” throughout the manuscript in the latest version of the manuscript.

2)A lot of work by researchers were listed and many organic substances promoting the chemical oxidative degradation of pollutants were enumerated in this manuscript. However, this manuscript looks like a report rather than a review. What were the roles of those organic substances, and how they can affect the reactions? The discussion about that in every section was unclear, please clarify.

Response: Thank you very much for your comments and suggestions to revised the manuscript. The beginning of each section describes the mechanism by which each organism promotes the reaction. As in Chapter 2, The organic small molecular substance is mainly prepared by strengthening metal redox cycle and activated H2O2 or PDS to construct advanced oxidation System promotes response; The Chapter 3, Organic complexing agents can effectively prevent the precipitation of transition metals in a non-acidic environment, and the formed coordination field often affects the redox characteristics of Fe2+/Fe3+, Thereby promoting the oxidation effect of the Fenton-like system; Chapter 4 begins by describing the advantages and mechanisms of MOF materials over simple mixing.

3)The manuscript needs to be revised by a native speaker. The grammatical problems and the tense used in the manuscript should be especially paid attention to.

Response: Thank you very much for your comments. We revised it.

4) Line 248: " H2O2/PDS"->" H2O2 or PDS "?

Response: Thank you very much for your comments. We revised it to H2O2 or PDS in line 100 of revised version.

5) Line 252 and 254: Pay attention to superscripts and subscripts.

Response: Thank you very much for your comments. We revised it.

Reviewer 2 Report

The mini-review discusses the effects of organic substances in oxidation processes like the Fenton reaction. It is discussed how radial formation is enhanced.

Most parts of the paper are very descriptive, just mentioning the main facts from the cited papers (115 in total). However, in a good review paper I would expect more explanations. I could mention many parts, where I would lie to read such more detailed explanations, for example in lines 212-214 when the effects of different R- and S-combinations are discussed.  
Quite often complexing capacities are discussed, but no measured values for that are given, e.g., line 224. More discussion of measured physicochemical values is required.

In Chapter 5 about MOFs, I would like to see some figures which might explain the advantages of this class of materials. Other porous materials (zeolites, ordered mesoporous oxides) might offer the chance as well for self-assembling organic ligand or ions (line 315). It must be better discussed, why especially MOFs are beneficial. Again, the observed results with MOFs are only listed but not qualified commented. In a review I also expect some judging so that the reader can see what is most prospective. For example, in Chapter 5.2 copper and cobalt ions are discussed, but it remains open whether they are preferable compared to iron or not.
Furthermore, mostly the kind of used MOF is not specified. However, the synthesis of some MOFs can be very complex and expensive – thus some MOF structures are preferable to others – this should be considered as well.

Even in Chapter 6 no clear statements by the authors are given, what in their opinion are the best routes to go. In contrast, some statements remain unclear, e.g., what do you mean with: “few practical application cases of new organics and MOFs materials” ? (line 390) The following lines (391-395) indicate some criticism and doubts, but they are also too less specified.

Several formal errors should be corrected, e.g., in line 69 there is no need to write “Tartaric” – it should be “tartaric” or in line 115 the charges of the Mn ions should be given as superscript.
In line 131 (headline 3.1) it should be “Humic” – the same capitalization of the first word is suggested for all the other sub-headlines in Chapter 3.  
In lines 217, 219, 252-254, 286, 289, 298, in Table 2 and line 321 subscripts and superscripts must be added or re-checked, respectively.
In line 267 "∙OH radicals" are meant.

In line 314, what do the authors mean with “attention from scholars at home and abroad”? In an international paper the term “at home” is misleading.

Author Response

Dear Editors and Reviewers:

Thank you for your letter and for the reviewers' comments concerning our manuscript entitled “How Organic Substances Promote the Chemical Oxidative Degradation of Pollutants: A Mini Review” (No.sustainability-1352288). Those comments are all valuable and very helpful for revising and improving our paper, as well as the important guiding significance to our researches. We have studied comments carefully and have made correction which we hope meet with approval. Besides, our current manuscript followed the journal formatting guidelines of Sustainability. Revised portion are marked in red throughout the revised manuscript. The main corrections in the paper and the Responses to the reviewer’s comments are as flowing:

NOTE: All the Page and Line numbers where revisions were made refer to the Manuscript and Highlight with marked changes (Manuscript_revised version).docx. The Manuscript_Clean Version was the same version of the Manuscript_revised version with cleaned from all the marks.

Responses to the Reviewer #2’s comments:

1) Most parts of the paper are very descriptive, just mentioning the main facts from the cited papers (115 in total). However, in a good review paper I would expect more explanations. I could mention many parts, where I would lie to read such more detailed explanations, for example in lines 212-214 when the effects of different R- and S-combinations are discussed.

Response: Thank you very much for your comments and suggestions to revised the manuscript. The application of typical substances has been given a supplementary explanation, and the R-\S- configuration characteristics are added in line 274-275 of revised version.

2) Quite often complexing capacities are discussed, but no measured values for that are given, e.g., line 224. More discussion of measured physicochemical values is required.

Response: Thank you very much for your comments and suggestions to revised the manuscript. The reaction conditions of references are supplemented and clarified. The structure of typical organic compounds is supplemented in Table 2.

3)In Chapter 5 about MOFs, I would like to see some figures which might explain the advantages of this class of materials. Other porous materials (zeolites, ordered mesoporous oxides) might offer the chance as well for self-assembling organic ligand or ions (line 315). It must be better discussed, why especially MOFs are beneficial. Again, the observed results with MOFs are only listed but not qualified commented. In a review I also expect some judging so that the reader can see what is most prospective. For example, in Chapter 5.2 copper and cobalt ions are discussed, but it remains open whether they are preferable compared to iron or not.

Response: Thank you very much for your comments and suggestions to revised the manuscript. We added Figure 4 to explain the structure and catalytic oxidation mechanism of typical MOF materials. Furthermore, we added the discussion of MOFs in Chapter 5. We also added the advantages of Fe, Cu and Co in line 343、358、370 of revised version.

4)Furthermore, mostly the kind of used MOF is not specified. However, the synthesis of some MOFs can be very complex and expensive – thus some MOF structures are preferable to others – this should be considered as well.

Response: Thank you very much for your comments and suggestions to revised the manuscript. This paper MOF material divided into three classes of iron base, copper, cobalt and mixed base, mainly introduces the catalytic degradation effect of different material, not too much attention to synthetic steps, and most of the literature did not list the cost of material synthesis. Currently, It only complements advantages of various materials, not from the perspective of synthetic difficulty and cost comparison.

5) Even in Chapter 6 no clear statements by the authors are given, what in their opinion are the best routes to go. In contrast, some statements remain unclear, e.g., what do you mean with: “few practical application cases of new organics and MOFs materials” ? (line 390) The following lines (391-395) indicate some criticism and doubts, but they are also too less specified.

Response: Thank you very much for your comments and suggestions to revised the manuscript. We revised “few practical application cases of new organics and MOFs materials” to “most organics and MOFs materials are applied only in the laboratories”.

6) Several formal errors should be corrected, e.g., in line 69 there is no need to write “Tartaric” – it should be “tartaric” or in line 115 the charges of the Mn ions should be given as superscript.

Response: Thank you very much for your comments. We revised it.

7) In line 131 (headline 3.1) it should be “Humic” – the same capitalization of the first word is suggested for all the other sub-headlines in Chapter 3.  

Response: Thank you very much for your comments. We revised it.

8) In lines 217, 219, 252-254, 286, 289, 298, in Table 2 and line 321 subscripts and superscripts must be added or re-checked, respectively.

Response: Thank you very much for your comments. We revised it.

9) In line 267 "∙OH radicals" are meant.

Response: Thank you very much for your comments. We revised it to •OH in line 118 of revised version.

10) In line 314, what do the authors mean with “attention from scholars at home and abroad”? In an international paper the term “at home” is misleading.

Response: Thank you very much for your comments. We revised it in text.

Round 2

Reviewer 2 Report

To my opinion the mini-review was significantly improved by the revision.

Some few things have still to be altered:

  • In line 159 – it must be [9,40] in the citation.
  • In the caption of Figure 4 just a number of the reference should be given – the whole citation must become a part of the References list.
  • Line 363: What does “kytog 99]”? I think that should be just [99] – and the reference must be given in the list of References.
  • List of References, line 63 ff. – The numbering must be checked – There is a number of references with all the number “1” – that must be wrong.
  • In general – all the References in the list must be checked for the style. Currently, very often not the correct style (as required by the journal) is used.

Author Response

Dear Editors and Reviewers:

Thank you for your letter and for the reviewers' comments concerning our manuscript entitled “How Organic Substances Promote the Chemical Oxidative Degradation of Pollutants: A Mini Review” (No.sustainability-1352288). Those comments are all valuable and very helpful for revising and improving our paper, as well as the important guiding significance to our researches. We have studied comments carefully and have made correction which we hope meet with approval. Besides, our current manuscript followed the journal formatting guidelines of Sustainability. Revised portion are marked in red throughout the revised manuscript. The main corrections in the paper and the Responses to the reviewer’s comments are as flowing:

NOTE: All the Page and Line numbers where revisions were made refer to the Manuscript and Highlight with marked changes (Manuscript_revised version).docx. The Manuscript_Clean Version was the same version of the Manuscript_revised version with cleaned from all the marks.

Responses to the Reviewer #1’s comments:

1)In line 159 – it must be [9,40] in the citation.

Response: Thank you very much for your comments. We revised it.

2)In the caption of Figure 4 just a number of the reference should be given – the whole citation must become a part of the References list.

Response: Thank you very much for your comments. We revised it.

3)Line 363: What does “kytog 99]”? I think that should be just [99] – and the reference must be given in the list of References.

Response: Thank you very much for your comments. We revised it.

4)List of References, line 63 ff. – The numbering must be checked – There is a number of references with all the number “1” – that must be wrong.

Response: Thank you very much for your comments. We revised it.

5)In general – all the References in the list must be checked for the style. Currently, very often not the correct style (as required by the journal) is used.

Response: Thank you very much for your comments. We revised it.
